# Contribution of Metabolomics to Multiple Sclerosis Diagnosis, Prognosis and Treatment

**DOI:** 10.3390/ijms222011112

**Published:** 2021-10-15

**Authors:** Marianna Gabriella Rispoli, Silvia Valentinuzzi, Giovanna De Luca, Piero Del Boccio, Luca Federici, Maria Di Ioia, Anna Digiovanni, Eleonora Agata Grasso, Valeria Pozzilli, Alessandro Villani, Antonio Maria Chiarelli, Marco Onofrj, Richard G. Wise, Damiana Pieragostino, Valentina Tomassini

**Affiliations:** 1Institute for Advanced Biomedical Technologies (ITAB), Department of Neurosciences, Imaging and Clinical Sciences, University “G. d’Annunzio” of Chieti-Pescara, 66100 Chieti, Italy; mariannarispoli92@gmail.com (M.G.R.); annadigiovanni93@gmail.com (A.D.); valeriapozzilli@yahoo.co.uk (V.P.); alessandro.villani@studenti.unich.it (A.V.); antonio.chiarelli@unich.it (A.M.C.); onofrj@unich.it (M.O.); richard.wise@unich.it (R.G.W.); 2Department of Neurology, “SS. Annunziata” University Hospital, 66100 Chieti, Italy; gio.deluca05@yahoo.com (G.D.L.); maria.diioia@unich.it (M.D.I.); 3Analytical Biochemistry and Proteomics Research Unit, Centre for Advanced Studies and Technology (CAST), University “G. d’Annunzio” of Chieti-Pescara, 66100 Chieti, Italy; silvia.valentinuzzi@unich.it (S.V.); piero.delboccio@unich.it (P.D.B.); lfederici@unich.it (L.F.); 4Department of Pharmacy, “G. d’Annunzio” University of Chieti-Pescara, 66100 Chieti, Italy; 5Department of Innovative Technologies in Medicine and Dentistry, University “G. d’Annunzio” of Chieti-Pescara, 66100 Chieti, Italy; 6Department of Paediatrics, “G. d’Annunzio” University of Chieti-Pescara, 66100 Chieti, Italy; elegrasso601@gmail.com

**Keywords:** metabolomics, multiple sclerosis, biomarkers, disease modifying treatment, MRI, biofluids

## Abstract

Metabolomics-based technologies map in vivo biochemical changes that may be used as early indicators of pathological abnormalities prior to the development of clinical symptoms in neurological conditions. Metabolomics may also reveal biochemical pathways implicated in tissue dysfunction and damage and thus assist in the development of novel targeted therapeutics for neuroinflammation and neurodegeneration. Metabolomics holds promise as a non-invasive, high-throughput and cost-effective tool for early diagnosis, follow-up and monitoring of treatment response in multiple sclerosis (MS), in combination with clinical and imaging measures. In this review, we offer evidence in support of the potential of metabolomics as a biomarker and drug discovery tool in MS. We also use pathway analysis of metabolites that are described as potential biomarkers in the literature of MS biofluids to identify the most promising molecules and upstream regulators, and show novel, still unexplored metabolic pathways, whose investigation may open novel avenues of research.

## 1. Introduction

The term metabolomics appeared for the first time in the early 2000s [1,2] to indicate the comprehensive analysis of the metabolome, which was first defined as the collective set of metabolites produced or present in biological systems at a given point in time [3]. Metabolomics aims to identify and quantify low molecular weight compounds (<1500 Da), i.e., metabolites, whose concentrations vary in relation to genetic and environmental perturbations [2,4]. Metabolomics identifies biologically meaningful patterns more easily than other omics sciences like genomics, transcriptomics or proteomics, since metabolites are far fewer than genes, RNA transcripts and proteins [5]. The human metabolome is thought to contain about 42,000 metabolites [6], although this may be an overestimate due to the presence of exogenous metabolites originating from diet, pharmacotherapy and gut microflora [7]. The metabolome is thought to offer a promising link between genotype and phenotype and metabolomics approaches have spread in many areas of biomedical, pharmaceutical and toxicological research [7]. Moreover, metabolites tend to be more conserved across species and insights from metabolomic studies in model organisms can be more readily translated to humans [5].

Currently, the main analytical techniques in metabolomics are proton nuclear magnetic resonance (1H NMR) spectroscopy, liquid chromatography-mass spectrometry (LC-ms), and gas chromatography-mass spectrometry (GC-ms). The required sample amount in NMR spectroscopy is about 300 μL, while it is only 10–30 μL in mass spectrometry-based analysis [8]. NMR spectroscopy identifies different compounds based on hydrogen (1H) magnetic resonant frequency [4]; this process helps scientists detect and distinguish all compounds which contain hydrogen [7]. It is a rapid, non-destructive and reproducible technique [9], which requires almost no sample pre-treatment [7]. For this reason, NMR experiments are unbiased to the chemical nature of the analytes [10]. However, NMR spectroscopy has a lower sensitivity, in terms of detection limits, than LC- and GC-ms (micromolar versus picomolar) [11]. Mass spectrometry, coupled with a separation technique (liquid or gas chromatography), determines the composition of a biofluid based on the mass-to-charge (*m*/*z*) ratio of charged particles [7]. GC-ms needs derivatization and longer sample preparation time than LC-ms [4]. Although LC-ms is emerging as the more predominant platform in metabolomics, no individual method is capable of detecting all metabolites [12].

Metabolomic studies are operated with targeted or untargeted analytical strategies [13]. Targeted methods study a limited number of predefined metabolites by comparing the biofluid spectrum of interest to a library of reference spectra of pure compounds, with a high level of precision and accuracy [6]. On the other hand, untargeted approaches attempt to profile all detectable metabolites in the sample, avoiding the need for a prior specific hypothesis on a particular set of metabolites [14]. Consequently, these studies are characterized by the generation of large amounts of complex data and require high performance bioinformatic tools for their analysis [15].

Due to the limited sensitivity of analytical techniques currently applied in metabolomic research and the complexity of biological samples, multivariate approaches like principal component analysis (PCA) or partial least squares-discriminant analysis (PLS-DA) are necessary to consider all detected metabolites at the same time [7] and account for their correlations [14]. These approaches overcome the limitations of traditional univariate statistical methods which ignore the strong correlation between multiple metabolites and cannot adjust for other known confounding factors [14].

Metabolomics is currently used to screen for inborn errors of metabolism, a group of monogenetic metabolic disorders that can be lethal or result in irreversible organ damage in new-borns, if not diagnosed and treated [16]. Another application of metabolomics is in the early diagnosis of tumours, given that cancer causes alterations in cellular metabolism [17]. Metabolomics has also the potential to be introduced in clinical practice to predict patients’ responses to different treatments [18]. For example, it would be helpful in oncology for monitoring biological response to treatment and for preventing unnecessary toxicity and long-term sequelae in non-responders to the selected treatment [19]. Apart from cancer research, metabolomics has been applied to better understand physiopathology and improve diagnosis, prognosis and therapeutic regimens of many other conditions, like respiratory, metabolic, neurological and even psychiatric diseases [5,20].

## 2. The Potential of Metabolomics in Multiple Sclerosis (MS)

Multiple sclerosis (MS) is a chronic immune-mediated disorder of the central nervous system (CNS), characterized by demyelination and simultaneous axonal and neuronal degeneration that occur from the earliest clinical stage of the disease [21]. It represents the most common cause of neurological disability in young adults in the Western world, affecting mostly people aged 20–40 years old with a female to male ratio of 3:1 in recent decades [22].

In MS, metabolomics could provide new insights into disease mechanisms by identifying altered metabolic pathways [23,24,25]. It could also help in finding biomarkers for early diagnosis [26,27], monitoring of disease progression [28,29,30,31] and treatment response [14,32,33]. The identification of a metabolic signature to distinguish patients with MS at an early stage from healthy controls or those with a clinical isolated syndrome (CIS) would be of great utility [34]. In the same way, identifying metabolites that anticipate disease progression as determined by clinical or imaging measures would help in monitoring disease and provide secondary outcomes in trials of progressive MS [35]. In fact, one of the unresolved issues of MS is the transition from the relapsing-remitting (RR) to the secondary progressive (SP) phase, in which neurodegeneration, axonal loss and brain atrophy can progress independently of peripheral immune cell infiltration and hypothetically due to a cascade of events, including oxidative stress, energy deficiency, ionic imbalance and failure of neuroprotective and regenerative mechanisms [36]. While RR-MS can be controlled by available disease-modifying therapies (DMTs), the treatment armamentarium for SP-MS is still very limited [37,38]. Another unmet need is the understanding of pathological mechanisms which underlie the primary progressive (PP) form of MS, that is characterized by gradual worsening of neurological disability from onset [39] and has limited therapeutic options [40]. Since response to treatment can be different in each patient and extremely difficult to predict, the discovery and validation of prognosis and predictive biomarkers of therapeutic response would be very valuable [35].

Metabolomics has a potentially important role in neurological diseases like MS, given the difficulties in accessing brain tissue in vivo [5]. Metabolomic analysis can use multiple biofluids, but cerebrospinal fluid (CSF) is most commonly used in MS research, due to its close proximity to the pathological tissue and its useful interface with the systemic circulation via the choroid plexus [11]. However, it is difficult to obtain from healthy subjects and many studies that use it as a sample source suffer from lack of a healthy control group [41,42,43]. Therefore, serum/plasma and less commonly urine [44,45] or tears [46] are other currently used specimens because obtaining them is less invasive.

## 3. Evidence from the Literature

### 3.1. The Potential of Metabolomics as a Diagnostic Tool

The identification of a metabolic profile which can rapidly differentiate subjects with MS, especially at an early stage, from healthy controls has always been of great interest in clinical practice [47]. As early as 1988, CSF and plasma samples from MS patients and controls were compared to detect differences in free amino acid levels. Indeed, increase or decrease in plasma levels of certain amino acids were attributed to the selective disruption of the amino acid transport systems [48], since the plasma concentration of an amino acid is generally proportional to the binding constant for its transport into the brain [49]. Apart from the blood concentration of amino acids, an additional local factor in the brain related to the MS condition was also implicated in explaining differences in the CSF concentration of amino acids [48].

Analysis of the urine metabolic profile also appeared capable of differentiating MS patients from healthy controls and from patients with other neurological diseases [44]. Since discriminant analysis of the chemical entities in the 0.5–3.5 ppm region of the NMR spectrum showed the main differences between MS and non-MS patients, they were proposed as likely candidate markers of inflammatory demyelination [44], although these results have not been further explored.

Metabolomic analysis was not only able to discriminate MS patients from healthy controls and subjects with other a priori known neurological diseases [14,26,42,50,51,52,53,54,55], but also offered biomarker profiles to predict diagnosis in a second cohort of patients whose diagnosis was not revealed, with a moderate sensitivity and specificity [29,56].

It seems that MS patients can be distinguished from healthy controls and discriminated according to their disease stage (RR, SP or PP) also using a “biomarker score”, i.e., a sum of the number of biomarkers having a concentration above the 95% percentile serum concentration in controls [57]. In particular, RR-MS patients showed biomarker scores lower than progressive MS patients, but there were no statistical differences among progressive forms of MS. The biomarker score positively correlated with disability (measured by the Expanded Disability Status Scale, EDSS) and parameters of neurodegeneration detected by MRI [57].

### 3.2. Biochemical Differences According to Disease Stage

Metabolomic research in MS is complicated by the clinical course of the disease: patients have different phenotypes at different time points, and their metabolic profile probably differs before and after a relapse [5]. However, one of the earliest studies did not show significant differences in the levels of CSF metabolites between MS patients with longstanding disease and those with short disease duration [41]. Significant differences were not detected even comparing CSF parameters between RR and chronic progressive (CP) forms of MS [23,58]. There was only a tendency towards lower levels of lactate and glutamine in the RR group compared to those of the CP [23]. The reduction in glutamine, which is produced only by glial cells [59], and the significant correlation between lactate and glutamine in the MS group, suggested a change in astrocytic metabolism of RR-MS patients as the most likely explanation of the alterations in the levels of these metabolites [23]. In MS patients, lower serum levels of glutamine have been associated with low levels of selenium as reported by Mehrpour M et al. [26]. The authors explained such a correlation considering that glutamine acts as a precursor to glutathione (GSH), GSH is a cofactor of glutathione-peroxidase (GSH-Px) and selenium is a prerequisite for the activity of GSH-Px that lowers the formation of reactive oxygen species (ROS) [60]. These results suggest that selenium may be a useful element in the treatment of MS [61].

In contrast, another study found a significant increase in CSF lactate/creatinine ratio in MS patients compared to controls only during clinical exacerbations of RR course. Moreover, pathological values of lactate in CSF were detected in 53% of patients with active plaques, but in none of those with chronic plaques [62]. Considering CIS patients, lactate was moderately increased only in those with active plaques [34], maybe due to an increased lactate production by immune cells. This evidence highlights how metabolomics may easily offer information on CSF composition, which reflects in vivo brain metabolic activity at different stages of MS disease, and how metabolomics may be consistent with clinical and radiological findings.

Some other studies found significant differences in metabolites levels between RR and CP forms [24,63]. In particular, SP-MS patients appeared to have consistently higher concentrations of lactate and polyol pathway metabolites in the CSF compared to RR-MS patients [24]. This finding argues against an increase of CSF lactate resulting from the glycolysis of activated leukocytes in inflammatory plaques and suggests instead an increase in lactate and polyol pathway metabolites related to mitochondrial dysfunction [24,57].

### 3.3. Insights into MS Molecular Mechanisms

Increased lactate and decreased citric acid cycle intermediates (citrate, oxaloacetate) in CSF indicates an enhanced reliance on glycolysis and diminished mitochondrial energy generation [50,64]. Therefore, CSF metabolic profile might be useful to target and monitor the effects of future therapies aimed at preventing MS disease progression by protecting mitochondria or boosting mitochondrial energy metabolism [24]. MS patients may suffer an imbalance between energy production and consumption, which could explain the increase in circulating oxypurines and plasma uridine, as indicators of the tissue energy crisis [63].

Investigating the biochemistry of MS in relation to its natural history, it appears that, compared to controls, RR-MS patients who have <13 years disease duration show elevated levels of anti-inflammatory gastrointestinal tract acids (GTAs) and normal levels of mitochondrial stress biomarkers (i.e., very long chain fatty acids (VLCFA) species)), whereas SP-MS patients have statistically similar levels of GTAs, elevated mitochondrial stress biomarkers and elevated plasmalogen ethanolamines. RR-MS subjects with ≥13 years exhibit metabolic profiles intermediate between short-duration RR-MS and SP-MS [27]. Elevation of VLCFA species is a consequence of mitochondrial β-oxidation deficiency and upregulation of peroxisomal activity [65]. Higher GTAs levels in early RR-MS and lower GTAs levels in SP-MS suggest a protective response in early RR phase, that eventually diminishes in the progressive phase [27].

There is also evidence of oxidative/nitrosative stress due to an increase in serum levels of nitrite, nitrate and malondialdehyde (MDA) and a decrease in serum levels of ascorbic acid [63,66]. MDA elevation indicates that ROS-mediated lipid peroxidation is more active in MS patients than controls [43,63]. Lipid peroxidation is considered pathogenically relevant because oxidation-derived aldehydes may act as structural modifiers of proteins [43]. Since no increase in lipid peroxidability or major changes in fatty acids (FAs) profiles have been observed when comparing MS and non-MS patients, lipid peroxidation may be mainly derived from increased free radical production [43]. Autoantibodies against lipoxidized proteins have been detected in CSF and increased significantly in MS patients [43]. Therefore, it has been hypothesized that they may contribute to MS pathogenesis [67].

Oxidative and nitrosative stress appears related to disability. Indeed, interleukin-10 (IL-10), tumor necrosis factor-α (TNF-α), interferon-γ (IFN-γ), advanced oxidation protein products and nitric oxide metabolites (NOx) were significantly higher in MS patients with a higher EDSS [68].

In the search for the mechanistic drivers of the switch from RR-MS to CP forms, the kynurenine pathway (KP) has emerged as a possible factor. The KP is the major route that breaks down tryptophan and the metabolites produced can have neurotoxic or neuroprotective effects [69]. Initially, the induction of the KP may be beneficial due to an immunomodulatory effect, which is reflected in longitudinal data showing elevated kynurenine/tryptophan ratio during maintenance of a stable EDSS in RR-MS [25]. However, chronic KP activation changes the excitotoxic balance due to increased quinolinic acid (QA) production. Indeed, QA can lead acutely to human neuronal death and chronically to neuronal dysfunction by different mechanisms, of which N-methyl-D-aspartate receptor (NMDA-R)-excitotoxicity is the best characterized [70]. It has been shown that QA production increases with disease severity and appears higher in the PP-MS subgroup [25].

In a search for pathophysiological differences between SP-SM and RR-MS phenotypes, eight biochemical pathways appeared differentially altered in the CSF: (1) aminoacyl-tRNA biosynthesis, (2) phenylalanine metabolism, (3) tryptophan metabolism, (4) valine, leucine and isoleucine biosynthesis, (5) pyrimidine metabolism, (6) nitrogen metabolism, (7) valine, leucine and isoleucine degradation and (8) purine metabolism [71]. Specifically, tryptophan metabolism achieved the second highest significance in the comparison between RR and SP-MS patients. As stated before, tryptophan is a precursor of kynurenate, which is generated through the deamination of kynurenine. Kynurenate appeared elevated in the CSF of SP relative to RR-MS patients, indicating that its levels change throughout the course of the disease [71]. Kynurenate has been identified as a neuroprotective agent and a reduction in its synthesis can lead to neurotoxicity [72]. Kynurenate showed an opposite trend in serum, where it appeared highest in the RR-MS group [25]. Lumbar CSF reflects a complex equilibrium of molecular transport between blood, CSF and brain tissue, which may explain these different observations.

A unique lipid signature is recognized both inside and outside the CNS of MS patients, which suggests that the breakdown of the blood-brain barrier (BBB) contributes to determining the metabolic profile. The most abundant lipids in plasma from MS patients were glycerophospholipids, mirroring what can be found in the CSF [53]. Since these lipids have been linked to a more severe disease [73] and lipid metabolism is influenced by oxidative stress and mitochondrial dysfunction [43], the observed lipid profile may be a degenerative signature rather than an inflammatory one [53].

To address whether body mass index (BMI) influences lipid profile, MS patients with normal BMI were compared to those with high BMI and healthy controls. Indeed, there were differences in particular in ceramides, triglycerides and diacylglycerols. High BMI caused increase in plasma levels of different ceramide species between MS patients and healthy controls. The destruction of myelin, which may be exacerbated in MS patients with high BMI, has been considered responsible for these differences [74]. In fact, the immune-mediated destruction of the myelin membrane, highly enriched in sphingomyelin, probably results in increased recycling of this molecule into ceramide, which is consistent with previously reported alterations of sphingolipids in post-mortem studies on MS brains [75]. Ceramide-induced changes in DNA methylation on anti-proliferative genes were identified as responsible for the increased monocytic cell counts in MS patients with high BMI. The negative correlation between brain MRI volume measurements and monocyte cell counts highlights the importance of monocytes in neurodegeneration [74].

### 3.4. Potential Biomarkers for MS and Patients’ Symptoms: Is There an Association?

In the attempt to correlate clinical symptoms and putative biomarkers, it was found that increased IFN-γ was associated with higher pyramidal symptoms, increased interleukin-6 (IL-6) was associated with sensitive symptoms and increased carbonyl protein and IL-10, but lower albumin levels, were associated with cerebellar symptoms [68]. These results suggest that different pathophysiological mechanisms may underpin the impairment of functional systems. Overall, it seems that disease progression is associated with a T helper 1 (↑ IFN-γ), monocytic and regulatory T cells (↑ IL-10) response and a concomitant down-regulation of T helper 2 (↓ interleukin-4) functions [68].

Another attempt to link candidate biomarkers for MS and clinical characteristics discovered positive correlations between the levels of two lysophosphatidylcholines (lysoPCs) and the Link index, as well as between the level of phosphatidylinositol (PI) and disease duration [76]. This suggests a role for PI in the broadening of neurodegeneration. In contrast, the intensity of lysophosphatidylinositol (LPI) showed a negative correlation with EDSS, thus implying a protective function [76].

Considering alternative biological fluids, tears appeared as an attractive source of biomarkers for MS, showing a typical acyl-carnitines signature and alterations in several putative lipid biomarkers [46], some of which are already found in CSF [77]. This suggests a molecular cross-talk between CSF and lacrimal fluid [46].

### 3.5. Integration of Metabolomics and Neuroimaging Data

During an attempt to correlate metabolic profile, structural imaging and neurocognitive data, a significant reduction in arginine plasma concentration was found in RR-MS patients; this was correlated to T1 holes, white matter lesions and performance of executive functions [78]. This reduction may derive from shunting arginine to produce nitric oxide (NO) via the inducible nitric oxide synthase (NOS) pathway; indeed urea, a breakdown product of arginine in the urea cycle, was not significantly different between MS patients and controls. Histidine, which was significantly decreased in RR-MS patients, was positively correlated with performance on the Brief Visuospatial Memory Test [78]. This may be due to a neuroprotective effect of histidine, similar to that exerted following ischemic damage in rats [79].

An integrative analytic approach which combined MRI variables, proteins and metabolite concentrations was able to distinguish SP-MS from RR-MS patients with high confidence and better than any single measure [28]. Among metabolites, indolepyruvate was higher in SP-MS compared to RR-MS patients. Since indolepyruvate acts as a direct precursor of kynurenic acid in the presence of free radicals [80], this finding supports the hypothesis of the central involvement of the KP in the mechanism of neurodegeneration [28]. Myelin basic protein, macrophage-derived chemokine and 5,6-dihydroxyprostaglandin were associated with worse disease progression in SP-MS patients, suggesting that they may be useful for monitoring disease course and treatment of SP-MS patients [28].

### 3.6. Integration of Metabolomics and Gene Expression Data

In order to find a connection between gene expression and the metabolome, the associations between whole-genome expression data and metabolites informative for MS status were explored. Human leukocyte antigen *(HLA)-DRB1* (a known MS-related gene) was not associated with any metabolite. The expression of several other *HLA* genes involved with antigen presentation was associated with acylcarnitine C14:1 [81], which is an intermediate of FAs oxidation and an indicator of mitochondrial function and energy metabolism [82]. It has been hypothesized that a change in class II *HLA* gene expression may reflect an activated immune environment promoting a transition in immune cell energy metabolism [81].

### 3.7. Evidence from the Use of Multivariate Approaches

Among multivariate prediction models differentiating women with MS from the corresponding controls, orthogonal partial least squares (OPLS) demonstrated the greatest potential by showing that C21 steroids, two conjugated α/β-reduced androstanes and estriol positively correlated with the presence of MS in women [83]. Systematically increased levels of these circulating steroids are known to attenuate neuroinflammation in MS [84], possibly affecting the balance between neuroprotection and excitotoxicity [83].

Considering the three trimesters of pregnancy and post-partum, PLS-DA showed an unambiguous separation between the four periods [85]. Sphingolipids and ceramides showed interesting alterations during pregnancy. Specifically, there was an accumulation of glucosylceramide C16 in the third trimester, which was associated with a low relapse rate [85]. This suggests a potential role for glucosylceramide C16 in managing such a disease and highlights how pregnancy induces a particular metabolic profile.

Using multivariate approaches like non-linear kernel function and PLS-DA, MS patients can be correctly classified according to their disease stage, for example CIS or confirmed MS [25,30,31,86].

Both PCA and PLS-DA were able to differentiate relapse-free patients during 2-year follow up from patients with relapses. They also distinguished between patients remaining with an EDSS <3.0 or those reaching an EDSS >4.5 after two years of follow-up [87].

Using a combination of NMR spectroscopy and PLS-DA, models showed a significant separation between RR and SP groups, while RR and PP groups appeared clearly separated albeit not significantly. By contrast, PP and SP groups showed a marked overlap, suggesting that a similar pathologic mechanism may underlie these phases [29]. LysoPC 20:0 was significantly lower in the PP-MS group compared to the RR-MS group and showed a strong association with the PP-MS disease course over time. Since plasma concentration and activity of phospholipase A_2_ (PLA_2_) were similar in PP-MS patients and controls, a possible explanation for lysoPC 20:0 depletion is a disease-specific functional impairment of brain PLA_2_ [31]. It remains to be elucidated if lysoPC 20:0 could serve as a surrogate marker for the extent of neurodegeneration and/or for differential diagnosis.

### 3.8. Differences between Multiple Sclerosis and Other Demyelinating Disorders

Multivariate approaches have also been used for discriminating MS from other demyelinating disorders [88,89,90]. In particular, a PLS-DA model showed how an MS group was characterized by a higher serum concentration of scyllo-inositol and glutamine, whereas serum from a neuromyelitis optica (NMO) group contained more acetate, glutamate, lactate and lysine [88]. A specific increase in the synthesis of scyllo-inositol or an enhanced release by the brain of MS patients might explain this difference. Scyllo-inositol appears able to discriminate NMO from MS, with a better sensitivity than that of anti-aquaporin 4 (AQP4) antibodies (95% versus 50–70%) despite a lower specificity (87.8% versus >90%). Scyllo-inositol and acetate concentrations did not specifically correlate with the anti-AQP4 status and thus there was no distinction between anti-AQP4 seropositive and seronegative patients in the models [88]. However, anti-myelin oligodendrocyte glycoprotein (MOG)-antibody disease showed a different pathophysiological profile, being characterized by an increase in formate and leucine and a decrease in myoinositol levels [90].

Due to the cellular toxicity in common across different types of autoimmune inflammatory disorders of the CNS, and considering patients with MS, neuromyelitis optica spectrum disorders (NMOSD) and idiopathic transverse myelitis, 1-monopalmitin and 1-monostearin were found significantly upregulated in all disease states compared to controls [89]. Another study found shared metabolic features between MS and NMOSD patients, in particular up-regulation of 2-hydroxybutyrate, acetone, formate and pyroglutamate and down-regulation of glucose and acetate relative to healthy controls, which may be related to altered energy metabolism and FAs biosynthesis in the brain [91]. However, a single model with multiple metabolite variables in coordinated regression with clinical characteristics, EDSS, oligoclonal bands and protein levels was able to discriminate the disease states from healthy controls [89].

As a possible consequence of altered energy metabolism, the lipoprotein population in anti-AQP4 positive NMOSD patients appeared skewed toward larger particles whereas in plasma of RR-MS patients the lipoprotein particles were smaller [90]. However, since lipoprotein modifications can also occur in relation to inflammation [92], perturbations in plasma lipoproteins may also derive from brain injury and inflammatory response [90]. Even urine metabolites, which drive the separation between MS patients, anti-AQP4 positive NMOSD patients and healthy controls, have been related to alterations in energy and FAs metabolism, mitochondrial activity and the gut microbiota [45].

### 3.9. The Potential of Metabolomics in Monitoring Treatment Effects

Metabolomics may be useful in monitoring the effects of therapeutic interventions. For example, it has been shown that vitamin D supplementation changed the levels of metabolites involved in redox homeostasis. These changes were only minimal in the MS group [14]. Therefore, a resistance to the metabolic effects of vitamin D supplementation has been hypothesized in MS patients.

Metabolomics has also been employed to detect differences in plasma metabolic profile before and after treatment with DMTs. For example, considering RR-MS patients starting interferon beta (IFN-β), there was a different distribution of baseline samples compared to those obtained during IFN-β exposure, particularly at 24 months of treatment. Ketone bodies, glutamate and methylmalonate levels significantly decreased during treatment, whereas tryptophan levels increased, especially in the first 6 months of therapy. Moreover, baseline metabolic profile differed between responders and non-responders. In particular, non-responders had increased baseline levels of ketone bodies, mainly produced in conditions of enhanced catabolism such as in an inflammatory one [32].

Metabolomics and machine learning have also been used to predict immunogenicity against IFN-β [93]. Patients who developed neutralizing anti-IFN antibodies (ADA) had a distinct metabolic response to IFN-β in the first 3 months, with several metabolites regulated in different ways. In particular, IFN-β appeared to have an enhanced lipid-lowering effect in ADA-positive patients. Serum lipids could contribute to ADA development by altering immune-cells’ lipid rafts. This is suggested by the different composition of lipid rafts between MS patients who became ADA-positive and those who remained ADA-negative. ADA status could be predicted based on metabolite concentration at 3 months using machine learning algorithms [93].

Also glatiramer acetate appeared to influence metabolic profile, especially in full responders to treatment. In these patients there was a trend toward the metabolic profile of healthy subjects after a year of treatment [33].

Therefore, metabolomics appears to be a promising, non-invasive tool to guide MS treatment with DMTs.

## 4. Pathway Analysis

Among the several pathways which seem to play a role in MS pathogenesis, those with the best evidence are: (1) mitochondrial dysfunction [50,64] which leads to energy deficiency, elevation of VLCFAs [27] and increase in circulating oxypurines and plasma uridine as indicators of tissue energy crisis [63]; (2) oxidative/nitrosative stress, which is responsible for ROS-mediated lipid peroxidation [63] and alteration of protein structures [43]; (3) kynurenine pathway [69], whose chronic activation leads to increased production of QA and excitotoxicity [25]. In order to sum up findings from the literature and gain more insights about metabolic alterations in MS, we performed a pathway analysis by including all the metabolites described as potential MS biomarkers in the literature. We aimed to identify the relevant relationships, functions and canonical pathways by using the metabolic information described in Appendix A as input data. We obtained these data by performing PubMed literature search, employing (metabolomics), (biomarkers) and (multiple sclerosis) as search terms. Review and original articles were considered in order to screen for new relevant references. By grouping together all the results from the 48 identified metabolomic studies in MS, we aimed to discover if unexplored pathways may have a role in MS pathogenesis.

Pathway analysis has emerged as an invaluable means for understanding data obtained from “omics” platforms and several software tools have been developed to support pathway analysis for genomic, proteomic and metabolomic studies [35]. Specifically, Ingenuity Pathway Analysis (IPA) (Qiagen, Hilden, Germany) can be used to identify pathways associated with genes, proteins and metabolites found in the “omics” analysis and allows the combination of genomic, proteomic and metabolomic results within the biological pathway-generating network, thus giving biological context to these data [94].

We performed core analysis in IPA by including metabolites from different biological fluids (CSF, blood, urine), which we empirically associated with a value of 10 when described upregulated in MS compared to controls, a value of 0.1 when described downregulated or a value of 1 when a conflicting or uncertain activation state had been described. The search for canonical pathways was performed to understand the molecular pathways in which metabolites from the dataset were involved. Differentially expressed metabolites were also categorized to related diseases and functions. The upstream regulator analysis allowed us to identify molecular regulators of metabolite expression, which could be transcription factors, cytokines, small RNAs, receptors, or pharmacological agents [95]. We considered only the upstream regulators associated with a reliable overlap *p*-value and activation z-score. The overlap *p*-value [96] is a useful statistical measure based on significant overlap between metabolites in the dataset and known targets regulated by the predicted transcriptional regulator and it is considered statistically significant with values < 0.05. The activation z-score [96] has the primary purpose of inferring the activation state of predicted transcriptional regulators by measuring the agreement between the metabolite expression within the dataset and what is expected from the literature. The overall predicted state of the upstream regulator can be significantly upregulated, with values > 2, or significantly downregulated, with values < −2. Ultimately, the metabolites from the dataset have been visualized in terms of networks, which are based on inter-metabolite connections. The assumption is that the more connected a metabolite, the more influence it has and the more important it is.

The first meta-analysis was performed by employing 145 metabolites and their ratios were obtained by comparing metabolite levels in MS patients and controls, without taking account of the specific biological fluid.

Although no canonical pathway turned out to be significantly upregulated or downregulated, several pathways (Figure 1) have been highlighted as being likely to play a role in MS:▪“Free radical scavenging”, highly activated through upregulation of superoxide and ROS production, which suggests that cells undergo oxidative stress that feeds demyelination and neuronal damage [97].▪“Small molecule biochemistry”, upregulated by an increased release of hormones, eicosanoids and neurotransmitters, a higher synthesis of ROS and NO, an increased concentration of glutamate and a metabolic switch toward glycolysis with reduced lipid catabolism and weak GSH synthesis.▪“Cell death and survival”, which refers to apoptosis of astrocytes and other unspecified neuronal cells in the CNS.▪“Cell-to-cell signalling”, remarked with the activation of macrophages, phagocytes and myeloid cells and concomitant reduced capacity of providing powerful responses to stress, thus supporting the inner link with the immune system in the pathogenesis of MS.▪“Metabolic disease”, which comes from metabolic features that concur to create a pathogenic phenotype similar to that of hepatic steatosis.

The diseases and functions analysis performed by IPA, considering the highlighted biological functions listed above, showed that the most significantly altered ones were “binding of hormones” and “uptake of lipids” as downregulated functions, and “release of hormones and eicosanoids”, “quantity of L-amino acids” and “concentration of L-glutamic acid” as upregulated pathways in MS. This means that metabolites from the dataset support an increased release of eicosanoids, key mediators of inflammation, and lead to highly neurotoxic concentrations of the excitatory neurotransmitter glutamate. The excitotoxicity of glutamate is driven by its accumulation and missed conversion into glutamine, that represents the main ammonia scavenger in the brain [98]. In this context, the glutamate/glutamine cycle must be seen as the pathway required not only for the synthesis of glutamate in the astrocytes, from which the inhibitory neurotransmitter gamma amino-butyric acid (GABA) will be produced, but also for the ammonia discharge that could otherwise disrupt neuronal integrity. The accumulation of glutamate in the brain produces four main consequences: (1) the hyperactivation of glutamate receptors and the exacerbated stimulation of their signalling pathways, that culminates in an excess of intracellular calcium by NMDA receptors, whose effects include increased NOS and other catabolic enzyme production [99]; (2) the activation of lymphocytes that triggers the further release of glutamate in the synaptic space [100], feeding both glutamate excitotoxicity and chronic inflammation in the brain; (3) the dysregulation of GABA synthesis, that is found to be downregulated in progressive MS [101]; (4) hyperammonaemia, with excitotoxicity similar to that of urea cycle disorders or hepatic diseases with hyperammonaemia, where the upregulation of glutamate occurs in the brain with the triggering of NMDA receptors [102].

The upstream analysis revealed that curcumin is a possible upstream regulator with inhibited state of activation (overlap *p*-value: 0.00068; activation z-score: −1.98), whereas huntingtin (HTT) (overlap *p*-value: 0.000000113; activation z-score: 1.964) and angiotensinogen (AGT) (overlap *p*-value: 0.0684; activation z-score: 1.972) have been highlighted as upregulated upstream.

The curcumin pathway, downregulated in MS, is typically involved in the antioxidant response to xenobiotics and stressful agents [103]. Therefore, a possible approach to contrast both inflammation and neurodegeneration in MS could be the add-on use of curcumin, which seems to act as a free radical scavenger capable of preventing neuronal damage in subarachnoid haemorrhage in mice [104]. Upregulated HTT may have a pleiotropic role both in neuroinflammation, by regulating macrophage function [105], and in prevention of neurodegeneration, being involved in neuronal function and survival [106].

From the analysis of the networks in Figure 2, HTT has been found again as a possible biomarker in MS supporting the definition of regulator given by the upstream analysis. PARP1 (poly (ADP-ribose) polymerase (1)) is a ubiquitous enzyme that mediates poly-ADP-ribosylation of proteins in several contexts and is well-known to be associated with xeroderma pigmentosum, a rare disorder characterized by photosensitivity, susceptibility to skin cancer and neurodegeneration [107]. Moreover, PARP1 has been demonstrated to be involved in inflammation and to be upregulated in MS lymphocytes [108], suggesting an important role in the pathogenesis of immune-linked neurodegenerative disorders. CCND1 (cyclin D1), a member of the cyclin family, is known to play important roles in the regulation of the cell cycle and has been shown to be overexpressed in cancer, including brain tumours such as glioblastoma and glioma [109]. NOS, the enzyme required to synthetize NO, provides an important mediator for brain cell communication under the positive pressure of glutamate in the CNS: its dysregulation underpins neurodegeneration, since NOS contributes both to cell death and neurotoxicity [110]. NOS is also tightly correlated to hypertension [111], thus supporting the upregulation of AGT revealed in the upstream analysis. NF-κB (nuclear factor kappa-light chain enhancer of activated B cells) is one of best described regulators of neuroinflammation in MS [112]. AMPK (AMP-activated protein kinase) generally deactivates anabolism in favour of oxidative reactions that increase the antioxidant pool supply. It has a neuroprotective role because it promotes autophagy and prevents the aggregation of undegraded proteins in neuronal cells [113]. However, its involvement in neuroinflammation arises from the evidence that prolonged activation of AMPK could exacerbate neuronal damage, despite the inhibition of NF-κB pathway [114]. Proinsulin is the precursor of insulin that seems to drive brain angiogenesis [115] by mechanisms involving the hypoxia-inducible factor (HIF) pathway. Akt (protein kinase B) is directly connected to autoimmunity through the activation of the PI3K/AKT/mTOR pathway [116], suggesting that its regulation should become a crucial target for therapeutic purposes in MS. ALB (albumin) plays self-contradictory roles in MS patients, because it both exasperates cytokine storm and is processed by pathologically expressed enzymes, also providing a score for BBB integrity [117]. HDL (high-density lipoprotein) and LDL (low-density lipoprotein) are involved in the pathogenesis and progression of MS because of their capacity to modulate immune responses [118]. Erk (extracellular-signal regulated kinase) participates in immunomodulation and has been proposed as a possible therapeutic target [119]. AHR (aryl hydrocarbon receptor) has recently been associated with interleukin-22, whose exact role in the pathogenesis of MS is still unknown [120]. Sod (superoxide dismutase) is a detoxicant enzyme that becomes important in the case of dysfunctional and impaired ROS metabolism occurring in neurodegenerative disorders such as MS [121]. WNT10B (wingless-type MMTV integration site family, member 10B) is canonically linked to oncogenesis but recent studies have demonstrated that the WNT/beta-catenin pathway drives neuroinflammation and demyelination processes in MS [122]. Dgk (diacylglycerol kinase) catalyses the synthesis of phosphatidic acid from diacylglycerol and is highly expressed in several conditions, e.g., inflammation, where its activity is deeply correlated with the cyclooxygenase family via crosstalk interactions [123]. Aconitase, an important enzyme involved in the Krebs cycle, has been found hyperactivated in MS patients relative to controls [124], suggesting alterations in mitochondrial aerobic metabolism. Since cytosolic aconitase is involved in iron homeostasis [125], it remains to be explored if aconitase may be involved in iron deposition within cerebral deep grey matter structures of MS patients [126]. EGFR (epidermal growth factor receptor) has been widely shown to be involved in inflammatory conditions, as demonstrated by the use of its inhibitors that show anti-inflammatory activity via NOS/NF-κB inactivation [127]. p70 S6k (ribosomal protein S6 kinase B1) regulates protein synthesis under the pressure of several triggers, e.g., EGF, and promotes [128] cell migration through the modulation of the actin cytoskeleton. ALT (alanine transaminase) is a biomarker of liver function which could be monitored as an ammonia scavenger, since hyperammonaemia is a shared condition between hepatic disorders and MS. LDH (lactate dehydrogenase) transforms pyruvate into lactate under hypoxic conditions, suggesting that hypoxia phenotype could occur in MS.

### 4.1. Cerebrospinal Fluid (CSF) Pathway Analysis

Eighty-six metabolites among those present in the total datasheet were identified in the CSF of MS patients. For this reason, they were selected to be uploaded in IPA for core analysis. The main biological functions highlighted by the CSF specific list support the metabolic switch occurring with neuronal death typically detected in MS:▪“Small molecule biochemistry” is activated through the upregulation of glutamine and other amino acid uptake, the increased concentration of neurotoxic glutamate and triacylglycerols, the release of eicosanoids and ROS, and the concomitant inhibition of nucleotides synthesis, neurotransmitter release and peroxidation of lipids.▪“Molecular transport”, “Lipid metabolism”, “Amino acid metabolism” appear activated as well.▪“Organismal injury and abnormalities” is upregulated by the overt neuroinflammation.▪“Cell death and survival” and “cell-to-cell signalling” are matched with neuronal death and inhibition of orexin neurons and other cells.

The diseases and functions analysis performed by IPA showed that the most altered pathways among those described above are: “Ion homeostasis of cells” as a downregulated network, “Molecules linked to movement disorders”, “Release of eicosanoid”, “Transport of molecules” and “Uptake of glutamine family amino acid” as upregulated networks. The dysregulation of ion homeostasis in the CSF indicates an intracellular ionic imbalance possibly driven by the excess of calcium as a result of glutamate-induced NMDA receptor activation. At the same time, CSF in MS is characterized by inflammation, as highlighted by the increased release of eicosanoids and the growing need for a glutamine pool, since glutamate is accumulating and must be converted in order to both remove ammonia and avoid excitotoxicity [98].

The search for “Canonical Pathways” revealed that “tRNA charging” is activated in the CSF during MS (overlap *p*-value: 1.06 · 10^−22^; activation z-score: 1.134), suggesting that protein synthesis is upregulated. No upstream regulators were found with significant values.

The metabolites from the CSF dataset have been visualized in terms of networks by merging those with the most significant number of covering metabolites (Figure 3). Some results are shared with the total matrix previously analysed (Sod, LDL, HDL, AMPK, Proinsulin, EGFR, p70 S6k, PARP1, AHR, ALT, NOS, Akt, Erk, NOS, HTT), confirming that MS involves biological networks related to cellular signalling, inflammation, immunomodulation, neurotoxicity and neurodegeneration. However, some other factors have emerged: (1) LEP (leptin) is an important energy regulator and has been recently associated with MS because patients have higher levels than controls, likely needed to modulate inflammation [129]; (2) CaMKII (Ca^2+^/calmodulin-dependent protein kinase II) belongs to the Ca^2+^/calmodulin-dependent protein kinase subfamily, so it is directly related to glutamatergic activities of synapses in the brain: studies conducted on pain and neuropathy in MS animal models have revealed that CaMKII is upregulated and probably linked to IL-17 pro-inflammatory activity [130]; (3) *SCNA* (sodium channel protein para) genes encodes for a presynaptic neuronal protein which mediates the voltage-dependent sodium permeability of excitable membranes and regulates seizure threshold in *Drosophila* [131]; (4) JNK1 and 2 (c-Jun protein kinase 1 and 2) belong to the mitogen-activated protein kinase (MAPK) family and their dysregulation has been associated with inflammatory and neurodegenerative conditions, although these isoforms are not typically found in the CNS [132]; (5) PEPCK (phosphoenolpyruvate carboxykinase) is an enzyme involved in gluconeogenesis; (6) Ca^2+^ is the most important intracellular mediator for ionotropic receptor (e.g., NMDAs) signalling in the CNS and its increased flux is responsible for excitotoxicity by glutamate; (7) Gsk3 (glycogen synthase kinase 3) is an enzyme involved in glycogen synthesis; (8) BAX (Bcl-2-associated X protein) is linked to cellular apoptosis; (9) IL-6 is a cytokine with both anti- and pro-inflammatory activity; (10) Pld (phospholipase D1) has been related to a number of diseases, such as diabetes, cancer, Alzheimer’s disease, thrombosis and hypertension, viral infection, but mostly MS [133]; (11) Creb (cAMP response element-binding protein) is involved in axonal regeneration, oxidative stress and neuroprotection, as shown by Fingolimod activity [134].

### 4.2. Blood Pathway Analysis

The blood matrix derives from considering all the metabolites described in serum and plasma, obtaining a list of 91 compounds for IPA analysis. The most highlighted biological functions found are listed below:▪“Small molecule biochemistry” is activated by the upregulation of steroid/amino acid metabolism, lipid release and NO synthesis and by the downregulation of lipid/hormone binding, oxidation of lipids and amino acid uptake.▪“Molecular transport” is highlighted by the upregulation of quantity of calcium and other metal ions.▪“Lipid metabolism”, “Amino acid metabolism”, “Cell signalling” and “Free radical scavenging” are activated as well.▪“Cell death and survival” is activated with the upregulation of cell viability and neuronal cell function and the downregulation of necrosis and cell death of CNS cells.▪“Haematological system” is activated through the upregulation of myeloid cells, phagocytes, leucocytes and blood cell activation.

From the analysis of Diseases and Functions of the biological pathways discussed above, “Cell death of tumour cell lines” and “Binding of hormones” appear significantly downregulated while “Release of hormone”, “Quantity of amino acids”, “Stimulation of cells”, “Ion homeostasis of cells”, “Generation of ROS” and “Survival of organism” appear significantly upregulated.

The upstream analysis revealed that Afatinib, Sirolimus, CPT1B (carnitine palmitoyltransferase 1B), IL-37 (interleukin-37) and LEP are downregulated in a significant manner, while GATA4 (GATA binding protein 4) is upregulated. Afatinib (overlap *p*-value: 2.17 · 10^−31^; activation z-score: −2.828) is an irreversible inhibitor of the ErbB family of tyrosine kinases through the binding to the EGFR [135]. Sirolimus (overlap *p*-value: 5.76 · 10^−17^; activation z-score: −2.538) has been already proposed in clinical trials to reduce mean plaque area size and EDSS of MS patients [136]. These findings on Afatinib and Sirolimus suggest their possible use in MS therapeutic approaches. CPT1B (overlap *p*-value: 4.93 · 10^−20^; activation z-score: −2.449) is the enzyme required for FAs conjugation to carnitines before their translocation to the internal mitochondrial matrix; its downregulation reflects an inhibited lipid metabolism. Previous studies demonstrated that blocking or lacking CPT1A in MS animal models reduced demyelination and production of pro-inflammatory cytokines [137], but nothing specific about CPT1B was demonstrated. IL-37 (overlap *p*-value: 6.93 · 10^−11^; activation z-score: −2) is an anti-inflammatory cytokine. LEP (overlap *p*-value: 5.61 · 10^−5^; activation z-score: −1.938) has been found upregulated in MS [138]. Our data from upstream analysis are inconsistent with the results from the only meta-analysis available in the Literature [129]. GATA4 (overlap *p*-value: 5.28 · 10^−7^; activation z-score: 2) is known to modulate embryogenesis, myocardial differentiation and testicular development: its involvement in autoimmune disorders, such as rheumatoid arthritis, has been considered since its overexpression drives angiogenesis and could be therapeutically targeted [134].

The analysis of the merged networks (Figure 4) revealed that the crucial nodes involve Creb, AP1 (activator protein 1), IL-1 (interleukin-1), IgG (immunoglobulin G) and others already seen in the previous network analysis (ALT, HTT, EGFR, proinsulin, Ldh, Pld, Akt, Sod, LDL, NOS, AMPK, ERK1/2, HDL, Gsk3, NF-κB, p70 S6k). (1) AP1 is thought to activate a systemic inflammatory phenotype in MS animal models [139]; (2) IL-1 is among the major interleukins expressed in MS and other neurodegenerative disorders [140]; (3) elevated IgG blood levels enhance inflammatory responses across the BBB and have been related to both MS disease state and demyelination [141].

### 4.3. Urine Pathway Analysis

This matrix was constructed considering only 11 metabolites derived from the urine of MS patients. The results corroborate the clinical importance of HTT as a potential biomarker to be considered for further studies.

Overall, these results demonstrate how the integration of data from independent datasets contributes to create a complex, yet coherent scenario that may help to elucidate molecular mechanisms underlying the pathophysiology of MS.

## 5. Conclusions and Future Perspectives

Metabolomics holds promise in MS as a non-invasive, high-throughput and cost-effective tool of biomarker analysis, capable of detecting the early stage of the disease and the switch towards the progressive phase, capable of contributing to establish prognosis, but also capable of monitoring treatment efficacy and predicting treatment response. Moreover, it may assist in the development of novel targeted therapeutics by revealing biochemical pathways implicated in neuroinflammation and neurodegeneration.

Until now, no biomarker has shown sufficient sensitivity and/or specificity to be used for population screening. Moreover, most of the reported biomarkers have not been verified in large groups of patients and controls [35]. The existing studies are heterogenous in terms of samples used (blood, CSF, tears or urine), acquisition methods and analysis of metabolomic data [11]. They generally include a limited number of patients and controls from single-centre studies, which may not accurately reflect the wider population. Errors due to preanalytical assessment and interindividual patient variation may explain some contrasting results. Therefore, there is a need for studies that include a larger number of individuals with good description of patients and control categories. Moreover, a standardized protocol for each of the biofluids of interest will be necessary to compare different studies and carry out metanalysis.

In order to achieve the goal of identifying clinically relevant metabolic biomarkers, beyond rigorous and standardized experiments, one must take into account several sources of interindividual variability, such as the patient’s diet, comorbidities, genetic predisposition to certain diseases like diabetes, medicines used, microflora, physical activity and other conditions that may ultimately influence the metabolic profile [142]. Indeed, metabolites identified may be associated with MS, but they may also be linked to differences in genetic background, epigenetic regulations or altered microbiota. Specifically, microbiota affects host immunity, alters the integrity and function of biological barriers and has a direct effect on several types of CNS-resident cells, so it has emerged as a point of interest for the ability to mediate or modify disease risk, including neurological disorders [143]. However, understanding the microbiota’s metabolic profile and how it affects the neurological disease’s course is still an unmet need.

Currently, patients with MS are treated according to clinical subtypes, i.e., CIS, RR, SP or PP. However, disease course varies greatly among patients belonging to the same clinical subtype and predicting disease trajectories from the beginning remains challenging [144]. Indeed, variable degrees of neuroinflammation and neurodegeneration affect relapse rate, disease progression and, overall, disease severity in the individual patient [145]. Therefore, some patients need an early intensive treatment, whereas others may initially benefit from a moderate-efficacy medication, eventually followed by escalation to higher-efficacy treatments [146]. Clinical and conventional MRI measures can capture only part of the pathological processes underlying MS [147] and other advanced MRI measures, such as diffusion tensor imaging or magnetization transfer ratio imaging, have the potential to detect neuropathological changes occurring in normal-appearing brain tissue. In this context, metabolomics may represent a complementary tool to identify pathogenetic processes that are relevant in each individual, to classify patients according to the underlying biology and, ultimately, to decide on appropriate, more targeted treatment.

Currently available DMTs target neuroinflammation [144], but are not able to prevent neurodegeneration. Metabolomics and other omics approaches may reveal biochemical alterations implicated in neurodegeneration such as mitochondrial dysfunction, oxidative/nitrosative stress, dysregulation of tryptophan metabolism, and thus contribute to the development of novel medications with neuroprotective effects. While no single omics analysis has the ability to capture the complexity of a living system [35], their integration should provide a more complete overview of the subject’s condition: for example, given an altered metabolic pathway, proteomics may highlight which enzymes are inactivated or stimulated and thus responsible for the dysregulation of certain metabolites. Considering a complex disease like MS, new targets for individualized treatment (e.g., enzymes) and related biomarkers for treatment monitoring (e.g., enzyme substrates) may emerge from this integration.

In conclusion, since MS relies upon complex biological mechanisms and different phenotypes contribute to its natural history, it is unlikely that a single metabolite exists that can distinguish a specific disease state or predict a treatment response. Therefore, it is probably more productive to search for sets of multiple metabolites, which may indicate an MS diagnosis, establish prognosis, reveal a transition towards the progressive phase and monitor treatment efficacy, possibly in association with clinical and radiological measures. Integrative analytic approaches, which associate clinical characteristics, MRI variables, protein and metabolite concentrations, appear able to distinguish MS subtypes better than any single measure [28]. Insights into pathophysiological mechanisms coming from the combination of neuroimaging-based diagnostics and omics could represent an important avenue of research. Therefore, future studies should consider not only metabolomic data, but also information from other omics approaches (e.g., genomics [148], epigenomics [149], transcriptomics [150] and proteomics [151]) and other modalities (e.g., MRI [152], neurophysiology, clinical scores [153]).

## Figures and Tables

**Figure 1 ijms-22-11112-f001:**
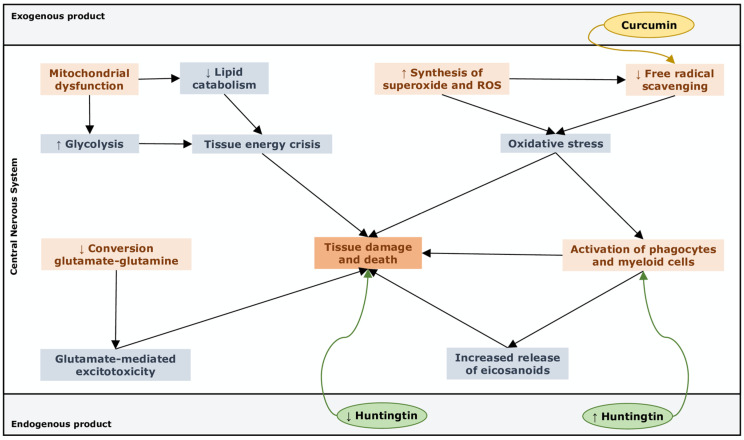
Orange boxes indicate pathophysiological triggers of tissue damage. Light cyan boxes describe metabolic pathways, in which metabolites from the dataset are involved. These metabolic pathways are involved in MS pathophysiology through alteration of biological functions such as small molecule biochemistry, controls of cell death, metabolic disease and cell to cell signalling. Dark orange boxes represent the endpoint of the metabolic cascades that lead to tissue damage and death. Upstream regulators, i.e., molecular regulator of metabolite expression, as identified by our pathway analysis, can promote tissue damage and death directly or through an activation of the immune system. These regulators and their effects are indicated in green (for huntingtin, an endogenous product) and in yellow (for curcumin, an exogenous product).

**Figure 2 ijms-22-11112-f002:**
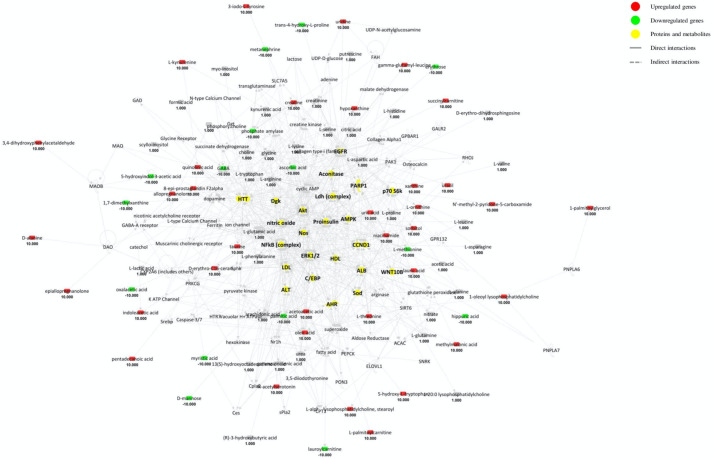
Merged networks based on inter-metabolite connections from the total matrix. Red and green shapes indicate genes significantly increased or decreased in expression in MS patients, whereas the number below represents the fold change log. The relationship between genes may lead to direct (solid lines) or indirect interaction (dashed lines). Yellow shapes indicate proteins and metabolites from the total matrix.

**Figure 3 ijms-22-11112-f003:**
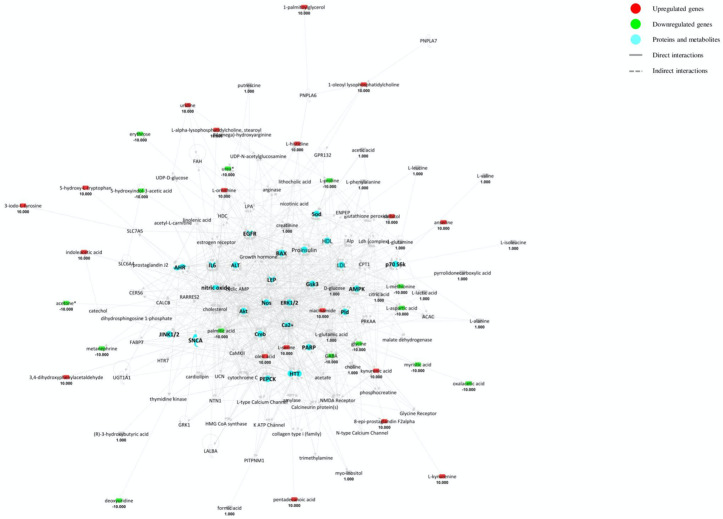
Merged networks based on inter-metabolite connections from the cerebrospinal fluid (CSF) matrix. Red and green shapes indicate genes significantly increased and decreased in expression in MS patients, whereas the number below represents the fold change log. The relationship between genes may lead to direct (solid lines) or indirect interaction (dashed lines). Light blue shapes indicate proteins and metabolites from the CSF matrix.

**Figure 4 ijms-22-11112-f004:**
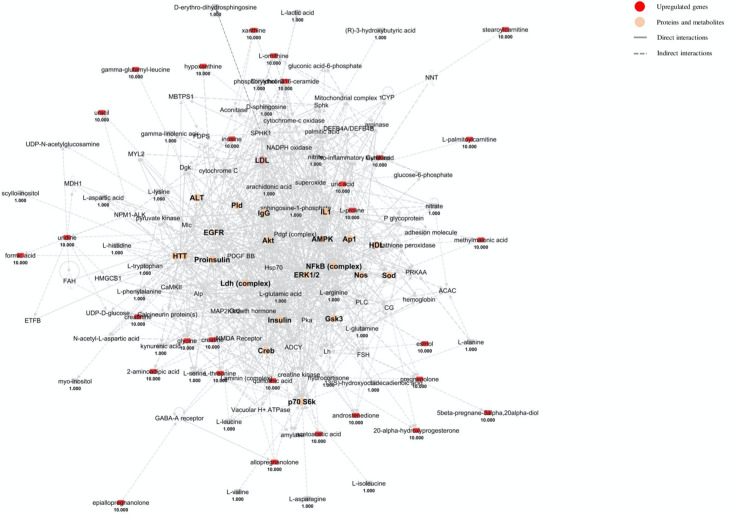
Merged networks based on inter-metabolite connections from the blood matrix. Red shapes indicate genes significantly decreased in expression in MS, whereas the number below represents the fold change log. The relationship between genes may lead to direct (solid lines) or indirect interaction (dashed lines). Light orange shapes indicate proteins and metabolites from the blood matrix.

## Data Availability

Not applicable.

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
