# Peer review of "Contribution of Metabolomics to Multiple Sclerosis Diagnosis, Prognosis and Treatment"

_ijms, 2021, doi:10.3390/ijms222011112_

Round 1
Reviewer 1 Report
The authors report on the state of the art in terms of metabolomic application to MS.
The article is very interesting, takes up recent advances in this field and distinguishes the putative role of the metabolomic biomarker in the diagnosis, the prognosis but also in the therapeutic response.
The article is very original and innovative and must be published.
In the NMO part, it is more appropriate to differentiate patients with AQP4-Ab, MOG-Ab and double negative rather than to consider the whole group of patients. Authors should extract this information from relevant articles.
Author Response
Reviewer #1
- The authors report on the state of the art in terms of metabolomic application to MS. The article is very interesting, takes up recent advances in this field and distinguishes the putative role of the metabolomic biomarker in the diagnosis, the prognosis but also in the therapeutic response. The article is very original and innovative and must be published. In the NMO part, it is more appropriate to differentiate patients with AQP4-Ab, MOG-Ab and double negative rather than to consider the whole group of patients. Authors should extract this information from relevant articles.
We thank the reviewer for his/her comments. As suggested, we specified the antibody status of NMOSD patients, where possible. Unfortunately, Park et al. (2016) and Kim et al. (2017) did not indicate the antibody status of their patients and thus this information is not identifiable from their articles. Therefore, evidence from their works refers to NMOSD patients (pp. 16-17):
"A specific increase in the synthesis of scyllo-inositol or an enhanced release by the brain of MS patients might explain this difference. Scyllo-inositol appears able to discriminate NMO from MS, with a better sensitivity than that of anti-aquaporin 4 (AQP4) antibodies (95% versus 50-70%) despite a lower specificity (87.8% versus > 90%). Scyllo-inositol and acetate concentrations did not specifically correlate with the anti-AQP4 status and thus there was no distinction between anti-AQP4 seropositive and seronegative patients in the models (88). However, anti-myelin oligodendrocyte glycoprotein (MOG)-antibody disease showed a different pathophysiological profile, being characterised by an increase in formate and leucine and a decrease in myoinositol levels (90).
Due to the cellular toxicity in common across different types of autoimmune inflammatory disorders of the CNS, and considering patients with MS, neuromyelitis optica spectrum disorders (NMOSD) and idiopathic transverse myelitis, 1-monopalmitin and 1-monostearin were found significantly upregulated in all disease states compared to controls (89). Another study found shared metabolic features between MS and NMOSD patients, in particular up-regulation of 2-hydroxybutyrate, acetone, formate and pyroglutamate and down-regulation of glucose and acetate relative to healthy controls, which may be related to altered energy metabolism and FAs biosynthesis in the brain (91). However, a single model with multiple metabolite variables in coordinated regression with clinical characteristics, EDSS, oligoclonal bands and protein levels was able to discriminate the disease states from healthy controls (89).
As a possible consequence of altered energy metabolism, the lipoprotein population in anti-AQP4 positive NMOSD patients appeared skewed towards larger particles, whereas in plasma of RR-MS patients the lipoprotein particles were smaller (90). However, since lipoprotein modifications can also occur in relation to inflammation (92), perturbations in plasma lipoproteins may also derive from brain injury and inflammatory response (90). Even urine metabolites, which drive the separation between MS patients, anti-AQP4 positive NMOSD patients and healthy controls, have been related to alterations in energy and FAs metabolism, mitochondrial activity and the gut microbiota (45).”

Reviewer 2 Report
Overall, the article, Contribution of metabolomics to multiple sclerosis diagnosis, prognosis, and treatment, highlights many accurate points on the benefits of metabolomics and provides a well-annotated summary of its potential use in neurologic diagnostics, especially when diagnosing Multiple Sclerosis (MS).
Overall, the paper thoroughly highlights the methodology of metabolomics and highlights how it will be used in diagnostic processes. The paper highlights the importance of how metabolomics differ by the patient and the efficiency of such methods in finding individualized treatment (expansion on this idea may be helpful). The paper also describes and provides some evidence as to how there are multiple studies that have used metabolomics to diagnose certain levels of MS (another topic that should be expanded upon). Finally, in terms of future endeavors and efforts, this paper positively describes the alternative studies and focuses on the ideas of other “omics,” (another topic which may be expanded on, hopefully talking about the benefits, similarities/differences, and limitations to these other processes). Ultimately, the overall paper presents a great topic of discussion.
Comments
In critique, there are a few edits and limitations to the paper, which should be elaborated upon before further acceptance, to ensure validity and precision in the topics discussed.
The authors need to create a table highlighting some experiments which have already utilized metabolic screening, maybe neuroimaging-based diagnostics, and potential pairs of diagnostics, utilizing systems like MRS, fMRI, and similar techniques. This table should highlight the study, the types of diagnostics used, the subjects, a short description of the metabolomics prevalence, and the level of significance if used for diagnosis.
(These papers in the fields of Traumatic Brain Injury may provide you with a platform to start the collection of studies):
https://pubmed.ncbi.nlm.nih.gov/25387616/ https://pubmed.ncbi.nlm.nih.gov/22438191/
You can also utilize many of the papers cited in the section, “Insights into MS molecular mechanisms,” to highlight these metabolic mechanisms. As stated throughout the sections of the article, it seems like many examples have already used these metabolomics methods for diagnostic/ observational purposes, and an organized table could help the readers see its prevalence and efficacy in diagnosis, over the years. These tables should highlight the validation of the results, p-values, and the overall methodologies for reaching these results.
In addition, create another table similar to the one proposed, which highlights the efficacy of diagnosing relapsing-remitting (RR) and secondary progressive (SP) MS through novel metabolomics. Many citations (as already presented) can be used however, strong significance (p-values) and validity should be presented in the tables as well, to ensure the validity of the claims. A table on MS diagnostics should act as the main evidence for the claim of the paper.
Finally, although briefly discussed within the paper, try to further the discussion on the novel therapeutics, individualized treatment, and metabolomics-based therapeutics. Further discuss how biochemical pathways implicate neuroinflammation, neurodegeneration, and various severity scores, personalized to specific people. Maybe highlight how this method can be valuable in the progression of personalized medicine and individualized treatment in the future.
Overall, the grammar and structure of the paper were well organized, however, the syntax of a few sentences were lengthy and progressed as run-on sentences. Limiting multiple ideas per sentence and having succinct ideas are important and something the editors should watch out for.
A few edited examples are as follows:
Edit Line 57-58
“NMR spectroscopy allows the identification of different compounds based on the resonant frequency of 1H in a magnetic field (4) and it is suitable for the detection of all the compounds which contain hydrogen atoms (7).”
TO à
“NMR spectroscopy identifies different compounds based on Hydrogen magnetic resonant frequency (4); this process helps scientists detect and distinguish all compounds which contain hydrogen (7).”
Remove Line 150-151 (due to run-on syntax and unclear demonstrative evidence)
“Other possible sources of lactate accumulation are glycolysis in and/or around emerging plaques, in an attempt to compensate for reduced adenosine triphosphate (ATP) production, and reduced clearance from CSF (even if there has been no demonstration of this) (34).”
Author Response
Reviewer #2
- Overall, the article, Contribution of metabolomics to multiple sclerosis diagnosis, prognosis, and treatment, highlights many accurate points on the benefits of metabolomics and provides a well-annotated summary of its potential use in neurologic diagnostics, especially when diagnosing Multiple Sclerosis (MS). Overall, the paper thoroughly highlights the methodology of metabolomics and highlights how it will be used in diagnostic processes. The paper highlights the importance of how metabolomics differs by the patient and the efficiency of such methods in finding individualized treatment (expansion on this idea may be helpful). The paper also describes and provides some evidence as to how there are multiple studies that have used metabolomics to diagnose certain levels of MS (another topic that should be expanded upon). Finally, in terms of future endeavors and efforts, this paper positively describes the alternative studies and focuses on the ideas of other “omics,” (another topic which may be expanded on, hopefully talking about the benefits, similarities/differences, and limitations to these other processes). Ultimately, the overall paper presents a great topic of discussion.
We thank the reviewer for his/her suggestions and provide a point-by-point answer to the Reviewer’s questions or comments below. Modified text in the body of the manuscript is indicated in bold and is underlined. We expanded on the importance of metabolomics in finding individualised treatments. We also expanded on the integration of metabolomics and other “omics” approaches and cited relevant papers on genomics, transcriptomics and proteomics in MS (pp. 34-35 and refs 148-151):
“While no single omics analysis has the ability to capture the complexity of a living system (35), their integration should provide a more complete overview of the subject’s condition: for example, given an altered metabolic pathway, proteomics may highlight which enzymes are inactivated or stimulated and thus responsible for the dysregulation of certain metabolites. Considering a complex disease like MS, new targets for individualized treatment (e.g., enzymes) and related biomarkers for treatment monitoring (e.g., enzyme substrates) may emerge from this integration.”
- In critique, there are a few edits and limitations to the paper, which should be elaborated upon before further acceptance, to ensure validity and precision in the topics discussed. The authors need to create a table highlighting some experiments which have already utilized metabolic screening, maybe neuroimaging-based diagnostics, and potential pairs of diagnostics, utilizing systems like MRS, fMRI, and similar techniques. This table should highlight the study, the types of diagnostics used, the subjects, a short description of the metabolomics prevalence, and the level of significance if used for diagnosis. These papers in the fields of Traumatic Brain Injury may provide you with a platform to start the collection of studies:
https://pubmed.ncbi.nlm.nih.gov/25387616/; https://pubmed.ncbi.nlm.nih.gov/22438191/. You can also utilise many of the papers cited in the section, “Insights into MS molecular mechanisms,” to highlight these metabolic mechanisms. As stated throughout the sections of the article, it seems like many examples have already used these metabolomics methods for diagnostic/observational purposes, and an organized table could help the readers see its prevalence and efficacy in diagnosis, over the years. These tables should highlight the validation of the results, p-values, and the overall methodologies for reaching these results.
Until now, only a few studies combined metabolomics and conventional MRI measures in order to gain insights into pathophysiological mechanisms or to improve diagnosis. These studies are indicated in the Table S1 with an asterisk. Unfortunately, studies which associate metabolomics and advanced magnetic resonance imaging like fMRI or MRS are currently lacking, to the best of our knowledge. We believe that insights into pathophysiological mechanisms coming from the combination of neuroimaging-based diagnostics and omics could represent an important avenue of research, as stated on p. 35:
“Integrative analytic approaches, which associate clinical characteristics, MRI variables, protein and metabolite concentrations, appear able to distinguish MS subtypes better than any single measure (28). Insights into pathophysiological mechanisms coming from the combination of neuroimaging-based diagnostics and omics could represent an important avenue of research.”
- In addition, create another table similar to the one proposed, which highlights the efficacy of diagnosing relapsing-remitting (RR) and secondary progressive (SP) MS through novel metabolomics. Many citations (as already presented) can be used however, strong significance (p-values) and validity should be presented in the tables as well, to ensure the validity of the claims. A table on MS diagnostics should act as the main evidence for the claim of the paper.
To highlight how metabolomics can contribute to MS diagnosis, prognosis and treatment monitoring, we summarized the results from relevant studies in the field in the Supplementary Table S1. In this table, we have now re-organised the information, by dividing papers into the following sections:
- Metabolic differences between MS patients and controls
- Metabolic differences among disease stages
- Metabolic differences between MS and other demyelinating diseases
- Monitoring of treatment response
We have also added p-values to support the claims.
- Finally, although briefly discussed within the paper, try to further the discussion on the novel therapeutics, individualized treatment, and metabolomics-based therapeutics. Further discuss how biochemical pathways implicate neuroinflammation, neurodegeneration, and various severity scores, personalized to specific people. Maybe highlight how this method can be valuable in the progression of personalized medicine and individualized treatment in the future.
We further explained how metabolomics may contribute to treatment decision according to alterations in pathophysiological mechanisms, as discussed below (p. 34):
“Currently, patients with MS are treated according to their clinical subtypes, i.e., CIS, RR, SP or PP. However, disease course varies greatly among patients belonging to the same clinical subtype and predicting disease trajectories from the beginning remains challenging (144). Indeed, variable degrees of neuroinflammation and neurodegeneration affect relapse rate, disease progression and, overall, disease severity in the individual patient (145). Therefore, some patients need an early intensive treatment, whereas others may initially benefit from a moderate-efficacy medication, eventually followed by escalation to higher-efficacy treatments (146). Clinical and conventional MRI measures can capture only part of the pathological processes underlying MS (147) and other advanced MRI measures, such as diffusion tensor imaging or magnetization transfer imaging, have the potential to detect neuropathological changes occurring in normal-appearing brain tissue. In this context, metabolomics may represent a complementary tool to identify pathogenetic processes that are relevant in each individual, to classify patients according to the underlying biology and, ultimately, to decide on appropriate, more targeted treatment.”
We have discussed further how metabolomics may be helpful in developing novel therapeutics, individualised treatment, also in combination with other omics approaches (p. 34):
“Currently available DMTs target neuroinflammation (144) but are not able to prevent neurodegeneration. Metabolomics and other omics approaches may reveal biochemical alterations implicated in neurodegeneration such as mitochondrial dysfunction, oxidative/nitrosative stress, dysregulation of tryptophan metabolism, and thus contribute to the development of novel medications with neuroprotective effects.”
- Overall, the grammar and structure of the paper were well organized. However, the syntax of a few sentences was lengthy and progressed as run-on sentences. Limiting multiple ideas per sentence and having succinct ideas are important and something the editors should watch out for.
A few edited examples are as follows:
Edit Line 57-58
“NMR spectroscopy allows the identification of different compounds based on the resonant frequency of 1H in a magnetic field (4) and it is suitable for the detection of all the compounds which contain hydrogen atoms (7).”
TO
“NMR spectroscopy identifies different compounds based on Hydrogen magnetic resonant frequency (4); this process helps scientists detect and distinguish all compounds which contain hydrogen (7).”
Remove Line 150-151 (due to run-on syntax and unclear demonstrative evidence)
“Other possible sources of lactate accumulation are glycolysis in and/or around emerging plaques, in an attempt to compensate for reduced adenosine triphosphate (ATP) production, and reduced clearance from CSF (even if there has been no demonstration of this) (34).”
We thank the Reviewer for his/her suggestions to improve the syntax of some sentences. We have made changes throughout the text to reflect the requested simplification. However, for simplifications requiring deletion of parts of the text we have not left them highlighted in order to avoid confusion. We have also simplified Figure 1, for ease of reading.
